# Prevention of Anti-HMGCR Immune-Mediated Necrotising Myopathy by C5 Complement Inhibition in a Humanised Mouse Model

**DOI:** 10.3390/biomedicines10082036

**Published:** 2022-08-20

**Authors:** Sarah Julien, Douangsone Vadysirisack, Camil Sayegh, Sharan Ragunathan, Yalan Tang, Emma Briand, Marion Carrette, Laetitia Jean, Rachid Zoubairi, Henri Gondé, Olivier Benveniste, Yves Allenbach, Laurent Drouot, Olivier Boyer

**Affiliations:** 1INSERM U1234, PAn’THER FOCIS Center of Excellence, Université de Rouen, F-76000 Rouen, France; 2UCB Pharma, Cambridge, 02140 MA, USA; 3Department of Internal Medicine and Clinical Immunology, Pitié-Salpêtrière University Hospital, Sorbonne Université, Inserm, U974, F-75013 Paris, France

**Keywords:** autoantibody, complement, HMGCR, immune-mediated necrotising myopathy, myositis

## Abstract

**Introduction**: immune-mediated necrotising myopathy (IMNM) is associated with pathogenic anti-signal recognition particle (SRP) or 3-hydroxy-3-methylglutaryl-CoA reductase (HMGCR) antibodies, at least partly through activation of the classical pathway of the complement. We evaluated zilucoplan, an investigational drug, and a macrocyclic peptide inhibitor of complement component 5 (C5), in humanized mouse models of IMNM. **Methods**: purified immunoglobulin G (IgG) from an anti-HMGCR^+^ IMNM patient was co-injected intraperitoneally with human complement in C57BL/6, C5-deficient B10 (C5^def^) and Rag2 deficient (Rag2^−/−^) mice. Zilucoplan was administered subcutaneously in a preventive or interventional paradigm, either injected daily throughout the duration of the experiment in C57BL/6 and C5^def^ mice or 8 days after disease induction in Rag2^−/−^ mice. **Results**: prophylactic administration of zilucoplan prevented muscle strength loss in C5^def^ mice (anti-HMGCR^+^ vs. anti-HMGCR^+^ + zilucoplan: *p* = 0.0289; control vs. anti-HMGCR^+^ + zilucoplan: *p* = 0.4634) and wild-type C57BL/6 (anti-HMGCR^+^ vs. anti-HMGCR^+^ + zilucoplan: *p* = 0.0002; control vs. anti-HMGCR^+^ + zilucoplan: *p* = 0.0939) with corresponding reduction in C5b-9 deposits on myofibres and number of regenerated myofibres. Interventional treatment of zilucoplan after disease induction reduced the complement deposits and number of regenerated myofibres in muscles of Rag2^−/−^ mice, although to a lesser extent. In this latter setting, C5 inhibition did not significantly ameliorate muscle strength. **Conclusion**: Early administration of zilucoplan prevents the onset of myopathy at the clinical and histological level in a humanized mouse model of IMNM.

## 1. Introduction

Immune-mediated necrotising myopathies (IMNM) belongs to idiopathic inflammatory myopathies (IIMs) which encompass a group of acquired muscle disorders, including dermatomyositis, inclusion body myositis, polymyositis and overlap myositis. IMNM are rare and severe diseases with symmetrical and proximal muscle weakness, elevated levels of creatine kinase reflecting the degree of muscle cytolysis and myogenic patterns in electromyography [1]. IMNM are a subgroup of IIMs distinguished by particular histological features: the presence of necrotic and regenerated myofibres, complement deposits on myofibres, paucicellular lymphocytic infiltrate and reexpression of major histocompatibility complex class I (MHC-I). In two thirds of patients, IMNM are associated with the presence of specific autoantibodies (aAbs) directed against Signal Recognition Particle (SRP) or 3-Hydroxy-3-MethylGlutaryl-Coenzyme A Reductase (HMGCR), and this association suggests the autoimmune nature of IMNM [2,3,4,5]. Anti-SRP recognises the 54 kDa subunit of the SRP complex localised on the surface of the endoplasmic reticulum (ER) and is implicated in targeting neo-synthesized protein from ER. HMGCR is an enzyme located at the membrane of ER and is involved in cholesterol biosynthesis. The reactivity of anti-HMGCR aAbs is directed against the C-terminal catalytic domain [6]. Titres of anti-SRP and anti-HMGCR aAbs are correlated with muscle strength and creatine kinase level [7,8]. In vitro, anti-SRP and anti-HMGCR aAbs induced muscle fibre atrophy, the production of inflammatory cytokines and the impairment of muscle regeneration [9,10]. We recently demonstrated that the passive transfer of purified human IgG obtained from anti-SRP^+^ and anti-HMGCR^+^ patients reduced muscle strength in C57BL/6 and Rag2^−/−^ mice. This pathogenic effect was markedly reduced in C3^−/−^ mice [11]. These data, together with the observation of the complement deposition on the sarcolemma, indicate that these aAbs are pathogenic and that complement activation is involved.

Therapeutic approaches for IMNM include corticosteroids and immunosuppressants, such as azathioprine and methotrexate, that may be combined with rituximab and/or intravenous immunoglobulins [12]. Although these treatments are effective, a high proportion of patients experience relapses, leading to irreversible muscle damage and disability.

Abnormal complement activity is associated with autoimmune diseases and specific therapeutic drugs targeting complement are used. For instance, in systemic lupus erythematosus (SLE), a disease characterized by a wide range of clinical manifestations such as cutaneous lesions, arthritis, renal involvement and hematologic disorders, and the presence of anti-DNA aAbs [13], anti-C5 antibody has been successfully used off-label in the setting of severe nephritis [14]. Moreover, patients with neuromyelitis optica caused by the action of complement-activating IgG aAbs to aquaporin 4 were successfully treated with the anti-C5 antibody [15].

The presumed role of the antibody-mediated activation of the complement provides a rationale to investigate therapeutics targeting the complement system in IMNM. Zilucoplan, an investigational drug, is a 15-amino acid macrocyclic peptide with high affinity and specificity to the human complement component 5 (C5). The binding of zilucoplan blocks the downstream assembly of the membrane attack complex (MAC; C5b-9) by (i) inhibiting the cleavage of C5 by the C5 convertase into C5a and C5b and (ii) binding to preformed C5b to sterically block interaction with C6, thereby inhibiting the formation of membrane pores and subsequent cell death.

The goal of the present study was to pre-clinically evaluate the C5 inhibitor zilucoplan in a humanized mouse model of IMNM and to compare its efficacy when used in a preventive manner or therapeutic setting.

## 2. Materials and Methods

### 2.1. Human Samples

Plasma was obtained after a first plasma exchange from a patient who fulfilled the consensus criteria for IMNM with the presence of anti-HMGCR aAb [16]. The plasma was collected after written informed consent was obtained. The study was approved by the Ile-de-France III ethics committee. The presence of the anti-HMGCR aAb was titrated by the addressable laser bead immuno-assay (ALBIA), as previously described [7,17]. The control was an anti-HMGCR aAb negative plasma from a healthy blood donor (HBD), obtained from the Etablissement Français du Sang (EFS, Bois-Guillaume, France).

### 2.2. Mice

Eight-week-old female C57BL/6 mice were purchased from JANVIER Labs, France. Rag2 deficient (Rag2^−/−^) C57BL/6 and C57BL/10 B10.D2-Hc0 H2d H2-T18c/oSnJ, also called C5^def^ mice, were purchased from Jackson Laboratories. The protocol for animal experimentation was approved by the institutional ethics committee n°054 under the reference 1415-201508071625487.

### 2.3. IgG Purification

Total IgG were purified from anti-HMGCR^+^ or healthy control plasma by automated protein-G affinity purification on an ÄKTA^TM^ Start chromatograph (GE Healthcare Life Science, Uppsala, Sweden), controlled, and quantified, as previously described [11]. All samples were stored at −80 °C until use. 

### 2.4. Passive Transfer of Human IgG into Mice

Eight- to 10-week-old female C57BL/6 wild type, Rag2^−/−^ or C5-deficient mice were divided into groups of 8 animals and received daily intraperitoneal injections of 2 mg of IgG purified from anti-HMGCR^+^ or healthy donor for 8 to 14 days. To reduce anti-human xenogeneic immune responses, C57BL/6 wild type and C5-deficient mice received a single dose of 300 mg/kg of cyclophosphamide (Sigma Aldrich L’lsle-d’Abeau Chesnes, France) at day-1.

### 2.5. Human Complement Supplementation

The human complement was fresh serum depleted in IgG as described above; the experiments were performed at 4 °C to avoid complement degradation and stored at −80 °C until use. After IgG depletion, the haemolytic complement activity (CH50) was measured with the MicroVue CH50 Eq EIA assay (Quidel, San Diego, CA, USA). The mice were supplemented by daily intraperitoneal injections of 200 µL of IgG-depleted complement human serum.

### 2.6. In Vitro Classical Pathway Complement-Mediated Hemolysis Assay and ELISA for Complement Components

Complement-mediated haemolysis and human C5 and split products C5a and sC5b9 levels were monitored from mouse serum before and after the injection of pooled complement human serum.

In a 384-well flat bottom plate, 1.25 × 10^7^ antibody-sensitized sheep red blood cells (Complement Technologies, Tyler, TX, USA) were incubated with 20% C5-deficient B10.D2/0Sn mouse serum in the presence of 1.5% of C5-depleted serum (Complement Technologies, Tyler, TX, USA) with GVB++ buffer (Complement Technologies, Tyler, TX, USA). The plates were then incubated at 37 °C for 1 h and centrifuged at 1000× *g* for 3 min, and the supernatants were transferred to a new plate. To evaluate red blood cell lysis, the optical density was measured at 412 nm (OD412).

Human C5 levels were assessed using a human C5 ELISA kit (Abnova, Taipei, Taiwan), in which mouse sera samples were diluted 400-fold. The absorbance at 450 nm was measured with a spectrophotometer (M4 reader, Molecular Devices, San Jose, CA, USA) and the concentration of samples was calculated from the absorption and standard curve. Human C5 split products of C5a and sC5b9 levels were assessed using the MicroVue Complement Multiplex kit (Quidel, San Diego, CA, USA), where mouse sera samples were diluted 5-fold. The developed ELISA plates were read using a Q-view Imager Pro (Quansys Biosciences, Logan, UT, USA).

### 2.7. Zilucoplan Administration

Zilucoplan was administered subcutaneously daily to C57BL/6 wild type, Rag2^−/−^ or C5-deficient mice receiving purified IgG (anti-HMGCR^+^ or control) and supplemented by IgG-depleted complement human serum. The dose of S.C. administration of zilucoplan was 10 mg/kg. The zilucoplan injection started at day 0 or 8, 30 min prior to the administration of the complement, and until the end of the experiment at day 7 or 14.

### 2.8. Mouse Muscle Strength Evaluation

Muscle strength was evaluated by measuring the in situ muscle contraction upon electrostimulation, and the analysis was performed on LabChart7 software (AD Instruments, Sidney, Australia), as previously described [11].

### 2.9. Histological Analysis

Mouse muscles were snap frozen into isopentane cooled in liquid nitrogen and cut at 8 μm. Cryosections were stained using hematoxylin and eosin (H & E). Regenerating myofibres defined as centronuclear cells were counted manually from a section of *gastrocnemius* of mice receiving control IgG or anti-HMGCR^+^ IgG. The number of regenerating fibres was related to the surface of the cryosection.

For immunofluorescence C5b-9 analysis, cryosections were incubated with rabbit anti-C5b-9 Ab (Abcam, Cambridge, UK), followed by AlexaFluor647-labeled anti-rabbit IgG Ab (Life Technologies, Carlsbad, CA, USA). For immunofluorescence foetal myosin heavy chain analysis, cryosections were incubated with mouse anti-myh3 Ab (Santa-Cruz, Dallas, TX, USA), followed by AlexaFluor647-labeled anti-mouse IgG Ab (Life Technologies, Carlsbad, CA, USA). For laminin staining, muscle sections were incubated with a rabbit anti-laminin Ab (Dako, Glostrup, Denmark) followed by AlexaFluor488-labeled anti-rabbit Ab (Life Technologies, Carlsbad, CA, USA), and counterstained with DAPI. The controls for staining specificity (omission of the primary antibody) were always negative.

Image acquisition was obtained by microscopy and the images were processed using ImageJ.

### 2.10. Statistical Analysis

Data were compared by the Mann–Whitney test using Prism 7 software (GraphPad, La Jolla, CA, USA). Data are presented as mean ± SD; a *p*-value < 0.05 was considered statistically significant and represented as * *p* < 0.05, ** *p* < 0.01 or *** *p* < 0.001.

## 3. Results

### 3.1. Zilucoplan Prevents Onset of the Disease in C5-Deficient Mice Supplemented with Human Complement

Zilucoplan inhibits human but not rodent C5 activation. Therefore, in order to investigate the role of C5 activation in experimental myopathy, we transferred IgG-depleted normal human serum as a source of complement to C5^def^ mice followed by the administration of purified anti-HMGCR^+^ or control IgG every two days. The recipients of the anti-HMGCR+ IgG were treated with a daily administration of zilucoplan or vehicle control (Figure 1A). Anti-HMGCR aAbs were detected in the sera of mice injected with purified anti-HMGCR^+^ IgG (Figure 1B). As expected, muscle strength was significantly decreased in anti-HMGCR^+^ mice compared with control mice (Figure 1C). Importantly, zilucoplan-treated mice did not develop muscle weakness and displayed muscle strength comparable to mice treated with control IgG (anti-HMGCR^+^ vs. anti-HMGCR^+^ + zilucoplan: *p* = 0.0289; control vs. anti-HMGCR^+^ + zilucoplan: *p* = 0.4634). The histological analysis of muscles from mice treated with zilucoplan highlighted fewer regenerated fibres (anti-HMGCR^+^ vs. anti-HMGCR^+^ + zilucoplan: *p*= 0.0401) (Figure 1D,I), necrotic cells (Figure 1H) and C5b-9 deposits than muscles from mice injected with anti-HMGCR^+^ IgG without zilucoplan (Figure 1J). Moreover, the serum complement activity was decreased in mice treated with zilucoplan, as shown using an ex vivo haemolysis functional assay (Figure 1E). Consistently, they had lower levels of C5a and C5b-9 in the serum (Figure 1F,J).

These data support that C5 activation in this model is driven by transferred human C5 and that the C5 blockade is effective in preventing muscle strength loss in C5^def^ mice when injected daily from day 0, with less complement deposition on myofibres and consequently less necrosis/regeneration. 

### 3.2. C57BL/6 Mice Prophylactically Treated with Zilucoplan Are Protected from Myopathy

To confirm the essential role of human C5 activation in anti-HMGCR^+^-driven myopathy, we repeated the same experiment as above using C57BL/6 recipient mice (Figure 2A). At day 7, we were able to detect high levels of circulating anti-HMGCR aAbs (Figure 2B). Mice receiving anti-HMGCR aAbs rapidly developed myopathy, as attested by a reduction of muscle strength (control vs. anti-HMGCR^+^: *p* = 0.0037) (Figure 2C). In contrast, muscle weakness did not develop in zilucoplan-treated mice, whose muscle strength remained similar to controls injected with purified IgG from healthy donors (anti-HMGCR^+^ vs. anti-HMGCR^+^ + zilucoplan: *p* = 0.0002; control vs. anti-HMGCR^+^ + zilucoplan: *p* = 0.0939). Histological analysis of gastrocnemius muscles from mice treated with zilucoplan revealed a statistically significant reduction of regenerated myofibres as quantified by the proportion of centronucleated fibres/mm^2^ (anti-HMGCR^+^ vs. anti-HMGCR^+^ + zilucoplan: *p* = 0.0476) (Figure 2D). We also observed that necrotic cells as well as regenerating cells expressing foetal myosin were less abundant in muscles of mice treated with zilucoplan (Figure 2E,F). Moreover, zilucoplan reduced C5b-9 deposits on the surface of myofibres (Figure 2G).

Together, these data confirm that C5 is required to induce myopathy and that the inhibition of human C5 activation is effective in preventing IMNM in this humanized mouse model.

### 3.3. Intervention with Zilucoplan Does Not Fully Normalize Muscle Strength Loss in Animals with Established IMNM Disease

We next evaluated the effect of zilucoplan in a model of established disease by administering daily zilucoplan 8 days after initial injection of anti-HMGCR^+^ IgG (Figure 3A). To support longer studies, we used Rag2^−/−^ mice to avoid xenogeneic immunisation against the human IgG and complement. Fifteen days after injection of anti-HMGCR^+^ purified IgG, high levels of anti-HMGCR^+^ IgG were detected in the sera of mice (Figure 3C). Muscle strength was reduced on days 7 and 15 in mice injected with anti-HMGCR^+^ IgG compared with mice injected with healthy control IgG (control vs. anti-HMGCR^+^: *p* = 0.0084 at day 7 and *p* = 0.0140 at day 15). On day 15, zilucoplan-treated animals exhibited a modest recovery of muscle strength compared with controls, but this was not statistically significant (control vs. anti-HMGCR^+^ + zilucoplan: *p* = 0.1893 and anti-HMGCR^+^ vs. anti-HMGCR^+^ + zilucoplan: *p* = 0.0650). Compared with mice receiving anti-HMGCR^+^ IgG, *gastrocmemius* muscles from mice injected with anti-HMGCR^+^ IgG and zilucoplan have no more necrotic (Figure 3D) or regenerated myofibres (Figure 3E) whereas less C5b-9 deposits were observed on the muscle cryosections of mice treated with zilucoplan (Figure 3F).

Overall, these observations indicate a central role of complement activation in IMNM induction. C5 inhibition was effective in preventing myopathy but did not demonstrate statistically significant clinical efficacy in mice when administered after disease onset.

## 4. Discussion

The anti-SRP and anti-HMGCR aAb titres correlate negatively with muscle strength and positively with serum creatine kinase levels in IMNM patients [7,8]. These specific aAbs are mainly of the complement-activating IgG1 isotype, i.e., found in 81% (17 of 21) of anti-SRP^+^ patients and in all (15 of 15) anti-HMGCR^+^ patients [7,17].

In a previous report, we demonstrated that the passive transfer of IgG purified from anti-SRP^+^ or anti-HMGCR^+^ patients induces myopathy in mice, since the C-terminal domains of human and mouse HMGCR and SRP54 share 98% and 100% homology, respectively, allowing human aAbs to react against their cognate murine target [11]. Moreover, in our mouse model of IMNM, myopathy is reduced upon passive transfer of an IMNM patient’s serum in C3^−/−^ compared to wild-type mice [11]. Together with the observation of MAC deposits on the sarcolemma, this provided a rationale for evaluating C5 inhibition as a treatment. 

In a randomized double-blind placebo-controlled phase II trial, zilucoplan (0.3 mg/kg), administered S.C. daily, yielded meaningful improvements in the treatment of acetylcholine-receptor autoantibody-positive myasthenia gravis [18]. Acetylcholine-receptor aAbs are mainly of IgG1 and IgG3 isotypes, i.e., able to activate the complement classical pathway upon the formation of immune complexes [19]. Moreover, in a multicentre, randomized, double-blind, placebo-controlled phase III trial, zilucoplan has shown efficacy in subjects with generalized myasthenia gravis (https://clinicaltrials.gov/ct2/show/NCT04115293; https://www.ucb.com/stories-media/Press-Releases/article/UCB-announces-positive-data-in-myasthenia-gravis-with-zilucoplan-phase-3-study-results; accessed date 16 August 2022).

Here, we harnessed the pathogenicity of human aAbs in mice to evaluate the C5 inhibitor zilucoplan in a mouse model of IMNM. In this model of IMNM, we demonstrated herein that zilucoplan injected concomitantly with aAbs was effective in preventing muscle weakness and myopathy in wild-type C57BL/6 mice as well as in C5^def^ mice supplemented with the human complement. Despite this preventive efficacy, zilucoplan injected 8 days after induction of myopathy did not statistically significantly ameliorate muscle strength loss in Rag2^−/−^ mice. This is in line with the recent findings of a phase 2, randomized, double-blind, placebo-controlled, multi-centre trial in patients with IMNM, where zilucoplan failed to meet the primary and secondary endpoints (https://www.ucb.com/stories-media/Press-Releases/article/UCB-s-zilucoplan-shows-no-relevant-effect-in-immune-mediated-necrotizing-myopathy-IMNM; accessed date 16 August 2022).

This discrepancy between the preventive and therapeutic effects of C5 inhibition with zilucoplan may be due to additional non-complement-dependent mechanisms that are likely involved in aAb pathogenicity. Indeed, incubation with plasma from anti-SRP+ and anti-HMGCR^+^ patients induced hypotrophic myotubes and myofibre atrophy [10]. This was accompanied by an increase in the transcription of genes encoding transcription factors involved in muscle atrophy, such as TRIM63 (MURF1) and F-box protein (atrogin 1). Inflammatory cytokines and ROS associated with muscle atrophy were also increased. In addition, anti-SRP and anti-HMCR aAbs reduced myotube formation by decreasing the secretion of cytokines involved in muscle regeneration. Muscle atrophy and inhibition of muscle regeneration are two mechanisms involved in the pathogenicity of IMNM aAbs, at least in vitro [10]. This phenomenon was independent of complement activation in the experimental conditions used. One may therefore suggest that the benefit of inhibiting complement activation in IMNM may be counterbalanced in vivo, at least in part, by these deleterious effects of Abs on muscle atrophy and the inhibition of regeneration.

As a limitation of this study, it should be reminded that each single mechanism for Ab pathogenicity may be more or less prominent at different time points of the disease and likely from one patient to another. In fact, in our previous study [11], we observed that anti-SRP^+^ or anti-HMGCR^+^ IgG purified from different patients was able to induce myopathy in mice. Here, anti-HMGCR^+^ IgG from only one IMNM patient was used. Another limitation of this study is that the short 7-day or 15-day period of IgG injection does not fully recapitulate the chronic aAb exposure of IMNM patients and our model is therefore biased toward early pathological events.

This work opens the perspectives that an early application of complement inhibition in the human disease course of IMNM may be beneficial. Indeed, we evidenced that the timing of complement inhibition in the disease course is of importance, i.e., Zilucoplan is effective in preventing myopathy if concomitantly applied with IgG inducing disease but less effective in already induced disease. Therefore, there would be a rationale for also pre-clinically evaluating the administration of the complement inhibitor in combination with other therapeutics.

## Figures and Tables

**Figure 1 biomedicines-10-02036-f001:**
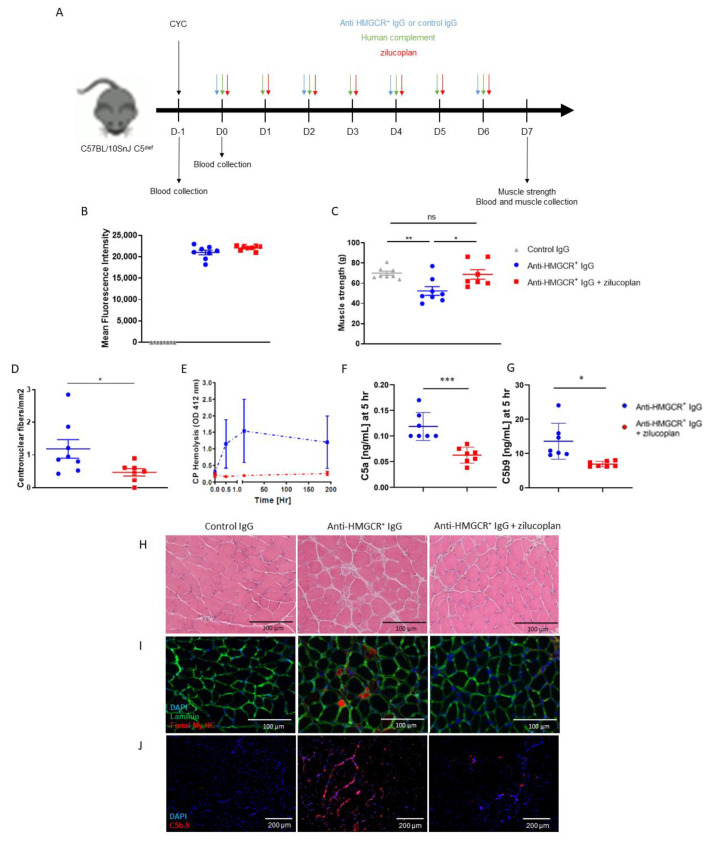
C5 inhibition prevents the development of IMNM in C5-deficient mice supplemented with human complement. (**A**) C57BL/10SnJ C5-deficient (C5^def^) mice (n = 8/group) were transiently immunosuppressed by a single injection of cyclophosphamide (CYC) and received injections of purified IgG from a patient suffering from anti-HMGCR^+^ IMNM with or without zilucoplan in conjunction with human complement. Control is IgG from normal serum in conjunction with human complement. (**B**) Circulating levels of anti-HMGCR aAb by addressable laser beads immunoassay (ALBIA) in the serum of mice at day 7. (**C**) Muscle strength was evaluated by measurement of gastrocnemius contraction upon electrostimulation (muscle strength). (**D**) Quantification of regenerative muscle fibres in all assessable frozen tibialis muscle biopsies from mice injected with control IgG or anti-HMGCR^+^ IgG with or without zilucoplan. (**E**) In vitro classical pathway complement-mediated haemolysis assay in serum from mice injected with control IgG or anti-HMGCR^+^ IgG with or without zilucoplan at day 0, 30 min, 5 h and day 7. (**F**,**G**) Quantification by ELISA of C5a and C5b-9 in serum from mice injected with control IgG or anti-HMGCR^+^ IgG with or without zilucoplan at day 0, 5 h. (**H**) H & E staining. Scale bar = 100 µm. (**I**) Immunodetection of regenerative myofibres after staining with mouse anti-foetal myosin heavy chain (My-HC) antibody followed by Alexa 647-labeled anti-mouse IgG antibody (red). Muscle fibres are made visible using rabbit anti-laminin antibody followed by Alexa 488-labeled anti-rabbit IgG (green) and DAPI (blue) for staining nuclei. Scale bar = 100 µm. (**J**) Complement deposits (red) on the surface of myofibres after staining with rabbit anti-C5b-9 antibody followed by Alexa 647-labeled anti-rabbit IgG and DAPI counterstaining (blue). Scale bar = 200 µm. Data are presented as mean ± SD; ns is for non-significant * *p* < 0.05, ** *p* < 0.01, *** *p* < 0.001 by Mann–Whitney two-tailed test. One out of two reproducible experiments is shown.

**Figure 2 biomedicines-10-02036-f002:**
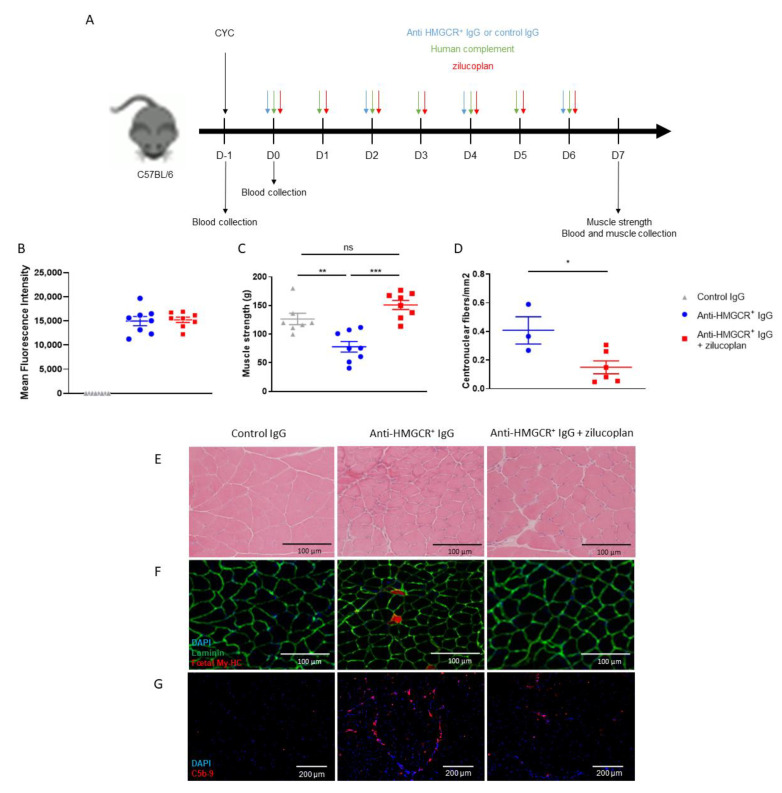
C5 inhibition prevents the development of IMNM in C57BL/6 mice supplemented with human complement. (**A**) C57BL/6 mice (n = 8/group) were transiently immunosuppressed by a single injection of cyclophosphamide (CYC) and received injections of purified IgG from a patient suffering from anti-HMGCR^+^ IMNM with or without zilucoplan in conjunction with human complement. Control is IgG from normal serum in conjunction with human complement. (**B**) Circulating levels of anti-HMGCR aAb by addressable laser beads immunoassay (ALBIA) in the serum of mice at day 7. (**C**) Muscle strength was evaluated by measurement of gastrocnemius contraction upon electrostimulation (muscle strength). (**D**) Quantification of regenerative muscle fibres in all assessable frozen tibialis muscle biopsies from mice injected with control IgG or anti-HMGCR^+^ IgG with or without zilucoplan. (**E**) H & E staining. Scale bar = 100 µm. (**F**) Immunodetection of regenerative myofibres after staining with mouse anti-foetal myosin heavy chain (My-HC) antibody followed by Alexa 647-labeled anti-mouse IgG antibody (red). Muscle fibres are made visible using rabbit anti-laminin antibody followed by Alexa 488-labeled anti-rabbit IgG (green) and DAPI (blue) for staining nuclei. Scale bar = 100 µm. (**G**) Complement deposits (red) on the surface of myofibres after staining with rabbit anti-C5b-9 antibody followed by Alexa 647-labeled anti-rabbit IgG and DAPI counterstaining (blue). Scale bar = 200 µm. Data are presented as mean ± SD; ns is for non-significant * *p* < 0.05, ** *p* < 0.01, *** *p* < 0.001 by Mann–Whitney two-tailed test. One out of two reproducible experiments is shown.

**Figure 3 biomedicines-10-02036-f003:**
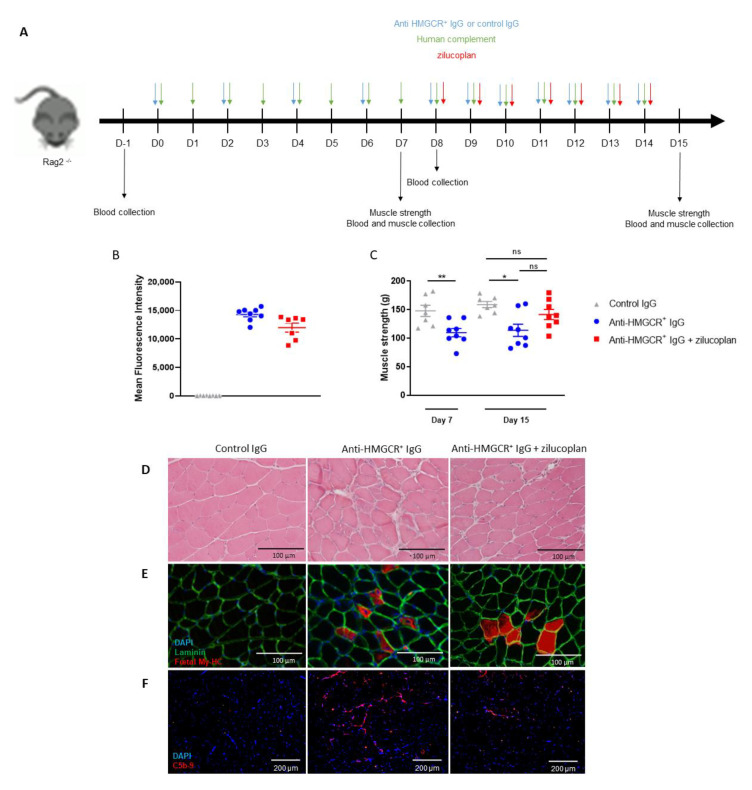
C5 inhibition partially ameliorates IMNM in Rag2^−/−^ mice supplemented with human complement. (**A**) Rag2^−/−^ mice (n = 8/group) received injections of purified IgG from a patient suffering from anti-HMGCR^+^ IMNM or purified IgG from normal serum in conjunction with human complement. Two groups were sacrificed at day 7 for assessing muscle strength. Moreover, three groups of Rag2^−/−^ mice received injections of IgG purified from normal serum or from a patient suffering from anti-HMGCR^+^ IMNM with or without zilucoplan daily from day 8 in conjunction with human complement and were sacrificed at day 15. (**B**) Circulating levels of anti-HMGCR aAb by addressable laser beads immunoassay (ALBIA) in the serum of mice at day 15. (**C**) Muscle strength was evaluated by measurement of gastrocnemius contraction upon electrostimulation (muscle strength). (**D**) H & E staining. Scale bar = 100µm. (**E**) Immunodetection of regenerative myofibres after staining with mouse anti-foetal myosin heavy chain (My-HC) antibody followed by Alexa 647-labeled anti-mouse IgG antibody (red). Muscle fibres are made visible using rabbit anti-laminin antibody followed by Alexa 488-labeled anti-rabbit IgG (green) and DAPI (blue) for staining nuclei. Scale bar = 100 µm. (**F**) Complement deposits (red) on the surface of myofibres after staining with rabbit anti-C5b-9 antibody followed by Alexa 647-labeled anti-rabbit IgG and DAPI counterstaining (blue). Scale bar = 200µm. Data are presented as mean ± SD; ns is for non-significant * *p* < 0.05, ** *p* < 0.01 by Mann–Whitney two-tailed test. One out of two reproducible experiments is shown.

## Data Availability

All data are available upon request.

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
