# Peer review of "Prevention of Anti-HMGCR Immune-Mediated Necrotising Myopathy by C5 Complement Inhibition in a Humanised Mouse Model"

_biomedicines, 2022, doi:10.3390/biomedicines10082036_

Round 1
Reviewer 1 Report
1) Abstract: Immune-mediated necrotising myopathy (IMNM) is associated with pathogenic anti-signal recognition particle (SRP) or 3-hydroxy-3-methylglutaryl-CoA reductase (HMGCR) antibodies, at least partly through activation of the classical pathway of complement. We evaluated zilucoplan, an investigational drug, and a macrocyclic peptide inhibitor of complement component 5 (C5), in humanized mouse models of IMNM. Purified immunoglobulin G (IgG) from an anti-HMGCR+ IMNM patient was co-injected intraperitoneally with human complement in C57BL/6, C5-deficient B10 (C5def) and Rag2 deficient (Rag2-/- ) mice. Zilucoplan was administered subcutaneously in a preventive or interventional paradigm, either injected daily throughout the duration of the experiment in C57BL/6 and C5def mice or 8 days after disease induction in Rag2-/- mice. Prophylactic administration of zilucoplan prevented muscle strength loss in C5def mice (anti-HMGCR+ vs anti-HMGCR+ + zilucoplan: p=0.0289; control vs anti-HMGCR+ + zilucoplan: p=0.4634) and wild-type C57BL/6 (antiHMGCR+ vs anti-HMGCR+ + zilucoplan: p=0.0002; control vs anti-HMGCR+ + zilucoplan: p=0.0939) with corresponding reduction in C5b-9 deposits on myofires and number of regenerated myofibres. Intervention treatment of zilucoplan after disease induction reduced complement deposits and regenerated myofibres in muscles of Rag2-/- mice, although to a lesser extent. In this latter setting, C5 inhibition did not significantly ameliorate muscle strength loss. Therefore, early administration of zilucoplan prevents onset of myopathy in a humanized mouse model of IMNM.
The abstract is quite rumbling and difficult to read. Could you please divide it in different sections (i.e. background, aim, methods, …)?
2) Abstract. Therefore, early administration of zilucoplan prevents onset of myopathy in a humanized mouse model of IMNM. Could you please underline the results that support this conclusion?
3) Abnormal complement activity is associated with autoimmune diseases and specific therapeutic drugs targeting complement are used. In systemic lupus erythematosus, antiC5 antibody has been successfully used off-label in the setting of severe lupus nephritis [13]. Could you please add a brief sentence on systemic lupus erythematosus and add these references:
A- Trabecular Bone Score and Bone Quality in Systemic Lupus Erythematosus Patients. Front Med (Lausanne). 2020 Sep 30;7:574842. doi: 10.3389/fmed.2020.574842.
B- Systemic lupus erythematosus: state of the art on clinical practice guidelines. RMD Open. 2018 Nov 27;4(2):e000793. doi: 10.1136/rmdopen-2018-000793.
4) Introduction. The presumed role of antibody-mediated activation of complement provides a rationale to investigate therapeutics targeting the complement system in IMNM. Zilucoplan, an investigational drug, is a 15-amino acid macrocyclic peptide with high affinity and specificity to human complement component 5 (C5). Binding of zilucoplan blocks downstream assembly of the membrane attack complex (MAC; C5b-9) by (i) inhibiting the cleavage of C5 by C5 convertase into C5a and C5b and (ii) binding to preformed C5b to sterically block interaction with C6, thereby inhibiting the formation of membrane pores and subsequent cell death. Here, we report on the evaluation of zilucoplan in a humanized mouse model of IMNM. Please improve the description of study aim.
5) 2.10. Statistical analysis Data were compared by the Mann-Whitney test using Prism 7 software (GraphPad)…. Please add some information in this paragraph (e.g. How the data will be presented?).
6) 3. Results 3.1. Zilucoplan prevents onset of the disease in C5-deficient mice supplemented with human complement. Please underline in the manuscript the most important statistical values to support the results.
7) 4. Discussion Anti-SRP and anti-HMGCR titres correlate with muscle strength and serum creatine kinase levels in IMNM patients [7,8]. These aAbs are mainly of the complement-activating IgG1 isotype and present in 81% (17 of 21) of anti-SRP+ patients and in all 15 anti-HMGCR+ patients [7,16]. Please improve the description of study results.
8) Please add a brief paragraph to report the limits of the study
9) This study indicates that the timing of complement inhibition in the disease course is of importance. Based on our results, zilucoplan is effective in preventing myopathy if concomitantly applied with IgG inducing disease. It is less effective in already induced disease. This opens the perspective that an early application of complement inhibition in the human disease course of IMNM may be beneficial. I would add a section for future perspectives in order to elaborate on your findings.
Reviewer 2 Report
the authors present a well-structured work with an excellent study methodology.
the results seem interesting, the iconography and histological preparations are adequate and functional to the aims of the study.
the results represent an interesting starting point for new therapeutic frontiers.
